# Assessing the Relationship among Land Transfer, Fertilizer Usage, and PM_2.5_ Pollution: Evidence from Rural China

**DOI:** 10.3390/ijerph19148387

**Published:** 2022-07-08

**Authors:** Lili Guo, Yuting Song, Mengqian Tang, Jinyang Tang, Bright Senyo Dogbe, Mengying Su, Houjian Li

**Affiliations:** 1College of Economics, Sichuan Agricultural University, Chengdu 611130, China; 14453@sicau.edu.cn (L.G.); 201907364@stu.sicau.edu.cn (Y.S.); tangmengqian@stu.sicau.edu.cn (M.T.); 201907298@stu.sicau.edu.cn (J.T.); 2018609002@stu.sicau.edu.cn (B.S.D.); 2College of Economics, Guangxi Minzu University, Nanning 530006, China

**Keywords:** land transfer, fertilizer usage, PM_2.5_ pollution

## Abstract

Concern for environmental issues is a crucial component in achieving the goal of sustainable development of humankind. Different countries face various challenges and difficulties in this process, which require unique solutions. This study investigated the relationship between land transfer, fertilizer usage, and PM_2.5_ pollution in rural China from 2000 to 2019, considering their essential roles in agricultural development and overall national welfare. A cross section dependence test, unit root test, and cointegration test, among other methods, were used to test the panel data. A Granger causality test was used to determine the causal relationship between variables, and an empirical analysis of the impulse response and variance decomposition was carried out. The results show that the use of chemical fertilizers had a significant positive impact on PM_2.5_ pollution, but the impact of land transfer on PM_2.5_ pollution was negative. In addition, land transfer can reduce the use of chemical fertilizers through economies of scale, thus reducing air pollution. More specifically, for every 1% increase in fertilizer usage, PM_2.5_ increased by 0.17%, and for every 1% increase in land transfer rate, PM_2.5_ decreased by about 0.07%. The study on the causal relationship between land transfer, fertilizer usage, and PM_2.5_ pollution in this paper is helpful for exploring environmental change—they are supplements and innovations which are based on previous studies and provide policy-makers with a basis and inspiration for decision-making.

## 1. Introduction

With the pursuit of high economic growth and a high standard of living worldwide, the environmental quality has been dramatically affected. Climate change and environmental pollution pose a serious threat to sustainable development and human health worldwide, which has aroused widespread concern [1]. In recent years, China’s air pollution crisis has become one of the most urgent environmental problems in China. China has experienced severe haze pollution, and the load of PM_2.5_ (particles with an aerodynamic diameter of less than *2.5* microns) is too high [2]. According to research by the World Health Organization (WHO), air pollution causes *800,000* deaths every year, among which PM_2.5_ produces the greatest influence on human health [3]. PM_2.5_ can absorb a large number of toxic substances because of its large surface area and high enrichment effect [4]. In many epidemiological studies, PM_2.5_ has been related to cerebrovascular, respiratory, and cardiovascular diseases [5]. Most researchers have found that long-term exposure to PM_2.5_ will negatively affect the heart and lungs [6]. For every *ten gm3* increase in PM_2.5_, respiratory mortality increases by *1.01%*, and cardiovascular diseases increases by *1.04%*. In addition, the rising speed of PM_2.5_ also leads to increases of *0.48%* and *0.60%* in the hospitalization rates of the respiratory system and cardiovascular diseases, respectively [3].

As a serious air pollutant, PM_2.5_ is mainly affected by the surrounding environmental conditions, industrial production activities, meteorological factors, and the excessive use of chemical fertilizers (nitrogen-containing components) in agricultural production and other human activities [5]. Some studies show that urbanization has an impact on the PM_2.5_ levels [7,8]. In addition, on the macroscale, meteorological conditions have been proven to have a considerable impact on PM_2.5_ pollution [3]. According to a study conducted in northern and western China [9], dust in spring/autumn will increase primary particles. PM_2.5_ pollution is also significantly related to land use patterns [10]. Li and Shen [11] believe that optimizing land use patterns at the city or community level is helpful for reducing PM_2.5_ pollution. 

In many parts of China, PM_2.5_ pollution is mainly affected by NH_3_ emissions [12], because secondary inorganic aerosols are the main component of PM_2.5_ [13]. As the only alkaline component in the atmosphere, NH_3_ can neutralize with sulfuric acid (H_2_SO_4_) and nitric acid (HNO_3_) in the atmosphere, producing a large number of secondary inorganic aerosols (the sum of sulfate, nitrate, and ammonium), causing severe haze pollution [14]. Moreover, ammonia from agricultural fertilizers is a major contributor to PM_2.5_ pollution around the world. Kawashima et al. (2022) found that ammonia from the use of agricultural fertilizers is a significant source of PM_2.5_ pollution [15]. In addition, Kang et al. (2022) found that ammonia produced by inorganic fertilizers and organic fertilizers had no significant differences in terms of soil impact [16]. 

At present, China has become the world’s largest emitter of NH_3_ [17]. China is a big agricultural country, and chemical fertilizers play an important role in global food production [18,19]. China’s fertilizer application has exceeded the economical optimal application rate [20]. Chemical fertilizer is one of the primary sources of atmospheric NH_3_ [21]. NH_3_ emissions from agricultural sources account for more than *80%* of the total NH_3_ emissions, including livestock and nitrogen fertilizer applications [22,23]. On average, only *30%* to *50%* of nitrogen is absorbed by crops [24], and a large amount of active nitrogen (Nr) is lost to the environment [25]. Nutrients that cannot be absorbed by crops seep into water or escape into the atmosphere, resulting in various environmental problems [26].

Controlling agricultural NH_3_ emissions has been proven to effectively reduce PM_2.5_ levels [27]. To reduce NH_3_ emissions from the source, on one hand, it is necessary to reduce the use of chemical fertilizers. On the other hand, it is necessary to improve the use efficiency of chemical fertilizers. As the market for the transfer of farmland rights continues to mature, the number of land transfers is increasing, and the number of large-scale farmers is also gradually increasing. Farmers with large farms are the main force in China’s future use of organic fertilizers, as they can pursue greater long-term agricultural benefits in this way [28]. Under the condition of reducing the number of applied chemical fertilizers, large-scale land management will not lead to a decline in output [29]. Meanwhile, for every 1% increase in farm scale, fertilizer use efficiency increases by 0.2%, reducing the environmental pollution caused by excessive use of chemical fertilizers [30].

The geographical and temporal distribution, source analysis, health consequences, and estimation of PM_2.5_ have been extensively studied by predecessors [31], and much valuable empirical evidence about PM_2.5_ pollution has been produced. However, few studies have paid attention to the relationship between land transfer, fertilizer usage, and PM_2.5_ pollution. Similarly, there is no consensus on the actual nature of their interaction. However, it is urgent and vital to investigate the relationship between land transfer, fertilizer usage, and PM_2.5_ pollution. Taking China as an example, this paper discusses the relationship between land transfer, fertilizer use, and PM_2.5_ concentration by using various econometric methods. 

Our research is particularly important for improving China’s air pollution and policy-making, and it has made scientific contributions in the following three aspects: Firstly, we used the panel vector autoregression (PVAR) model to reflect the heterogeneous influence of land transfer and fertilizer usage. As far as we know, this was the first time in China that the PVAR method has been used to study the relationship between land transfer, fertilizer usage, and PM_2.5_ pollution. The method helps to determine the direction of causality among PM_2.5_ air pollution, land transfer rate, and fertilizer use and helps to identify changes in the short-term and long-term effects between variables. Secondly, PM_2.5_ seriously threatens human life and health. Exploring the important sources of PM_2.5_ and putting forward targeted and effective policies to reduce PM_2.5_ pollution is of great significance. Finally, the Chinese government intends to achieve peak carbon emissions in 2030 and achieve carbon neutrality in 2060. China’s situation is attractive to the world. This paper is one of the few studies that has attracted international attention to key issues such as land transfer, fertilizer usage, and PM_2.5_ pollution in China.

The rest of this paper is organized as follows: Section 2 consists of a literature review and the construction of the research hypotheses; Section 3 includes the data sources and econometric methods; Section 4 introduces and discusses the empirical results; and the last section includes a summary and policy suggestions.

## 2. Literature Review and Research Hypotheses Construction

### 2.1. Land Transfer and Fertilizer Usage

For the relationship between land transfer and fertilizer usage, we put forward the following assumption:

**H1.** 
*Land transfer has a significant negative impact on the use of chemical fertilizers.*


The basis for our hypothesis is as follows:

Rural labor migration can effectively influence rural land circulation [32], and the role of migrant workers in promoting rural land circulation will increase over time [33]. Because of the reform and opening up, China has experienced a rapid and extensive urbanization process. According to the latest data from the seventh census, by the end of 2020, China’s rural population decreased to *509.79* million, the urban population increased to *901.99* million, and the urbanization rate had risen to *63.89%* [34]. High-speed urbanization has led to abandoned farmlands and rural labor loss in China [35], which has caused severe land-use problems [36]. In order to make full use of abandoned land in rural areas, the Chinese central government has also formulated relevant policies to promote rural land circulation, improve the land utilization rates, and encourage the circulation of agricultural land management rights [37]. In 2014, the General Office of the Communist Party of China (CPC) Central Committee and the General Office of the State Council issued the Opinions on Guiding the Orderly Circulation of Rural Land Management Rights to Develop Moderate Scale Operation of Agriculture, requiring that land circulation and moderate-scale operations should be vigorously developed, and the contractual management rights should be confirmed within five years. The reform of the “separation of powers”, put forward in 2016, enables those farmers who do not want to continue to engage in agriculture to trade their rural land use rights, and also enables those farmers who want to expand the scale of agricultural operations to lease rural land [38]. This kind of agricultural land use rights trade is usually called “rural land transfer” (RLT).

In recent years, the process of land transfer has been dramatically accelerated. The research shows that farmers prefer to use chemical fertilizers in order to reduce labor input under small-scale land production [39]. Land circulation expands the scale of rural agricultural land management, and the proportion of large-scale farmers in agricultural production will continue to increase in the future, becoming larger and larger [40]. Therefore, generally speaking, land circulation reduces chemical fertilizers and uses organic fertilizers instead. Due to the scale effect, large-scale farmers place more emphasis on future income and are more likely to improve land quality through the use of organic fertilizers [41]. Large-scale farmers have a higher socioeconomic status than small-scale farmers [42], requiring a higher reputation and stimulating them to apply organic fertilizer [43].

### 2.2. Fertilizer Use and Inhalable Particulate Pollution

For the relationship between fertilizer use and inhalable particulate pollution, we put forward the following assumption:

**H2.** 
*The use of chemical fertilizers has a significant positive effect on the concentration of PM2.5.*


The basis for our hypothesis is as follows:

Feeding the growing and increasingly affluent global population is arduous [39]. To meet this challenge, about two hundred tons of chemical fertilizers and three tons of pesticides have been used in global agricultural output [44]. China is the largest consumer of agricultural chemicals in the world. However, only *9%* of the world’s arable land consumes more than *30%* of chemical fertilizers and pesticides [45]. In the past 30 years, synthetic nitrogen fertilizer has played a vital role in ensuring China’s food security. However, low efficiency and a high percentage of agricultural chemical losses are widespread, resulting in economic losses and serious local, regional, and even global pollution [45]. 

The chemical formula of ammonia is NH_3_. High concentrations of NH_3_ from agriculture contributes significantly to PM_2.5_ pollution in China, with fertilization being the most crucial agricultural source of NH_3_ and PM_2.5_ in the atmosphere. The main particulate pollutants are produced when tractors disturb the wind, or the fertilizer particles are entrained in the soil. In addition, the photochemical reaction between ammonia and nitrogen oxides after fertilization produce secondary organic agricultural machinery aerosols and NH_3_ [46]. NH_3_ plays an underestimated role in the formation of the main components of the concentration of fine particulate matter (PM_2.5_) in Chinese cities. It has been found that in the main fertilization season (summer) and the occasional fertilization season (winter), the influence of NH_3_ emissions on the monthly average concentration change of PM_2.5_ in the first and second half of the year is *5.5* times and *1.5* times that of SO_2_ and NOx, respectively, which is a much greater factor than that of SO_2_ and NOx [12]. Since 1960, with the application of fertilizers, the global ammonia concentration has been increasing significantly [47].

### 2.3. Land Transfer and PM_2.5_ Pollution

For the relationship between land transfer and PM_2.5_, we put forward the following assumption:

**H3.** 
*The improvement of the land transfer is helpful to reduce PM_2.5_ pollution.*


The basis for our hypothesis is as follows:

In the past 20 years, more and more farmers are engaged in land transfer (leasing other farmers’ land) to expand the scale of the farmland they manage in China. According to data from the Ministry of Agriculture and Rural Affairs, the area of cultivated land in circulation increased from *4.25* million hectares to *30.67* million hectares between 2007 and 2016, and the proportion of cultivated land in circulation increased from *5.2%* to *35%* of the total cultivated land [11]. The transfer of land rights often leads to changes in land use and intensity, which inevitably leads to changes in regional carbon emissions [48]. The concentration of PM_2.5_ is similar to the growth trend of carbon emissions and has a spatial correlation. The changes in land use and land use intensity caused by land transfer may have an impact on the environmental problems of PM_2.5_ pollution. According to the existing research, the transfer of land rights can strengthen environmental protection and reduce air pollution [49,50]. First of all, the government’s stable transfer of land management rights ensures that large-scale farmers can obtain corresponding future benefits from the application of organic fertilizers. Their application of organic fertilizers has increased significantly [41], reducing ammonia emissions from synthetic fertilizers. Secondly, controlling ammonia emissions can effectively reduce PM_2.5_ [51]. Improving the technology and management of agricultural production can reduce agricultural NH_3_ emissions [52]. The extensive application of many technologies, such as covering and storing manure, replacing urea with animal manure, deep application of low ammonia, etc. [51], and improving extensive agricultural management [53] have great potential for reducing NH_3_ emissions, and the farm has been expanded by leasing land. Finally, the transfer of land increased the fragmentation of land—the number increased, but the location was not concentrated. According to the monitoring data of Chinese family farms in 2018, large-scale farmers managed an average of 15 scattered plots, and some farmers even managed as many as *1269* scattered plots [54]. Farmland fragmentation helped to reduce PM_2.5_ [29].

## 3. Materials and Methods

### 3.1. Variables and Data

The regional scope of the data selection is China’s 30 provincial-level administrative units, excluding Hong Kong, Macao, Taiwan, and Tibet. Due to different statistical calibers, we did not include data from Hong Kong, Macao, and Taiwan. Due to factors such as Tibet’s economy, education, and population, the previous data were missing, so the data area we selected did not include Tibet. Land turnover rate, fertilizer use intensity, and PM_2.5_ concentration are the three main variables in this study, among which land turnover rate = total household contracted cultivated land turnover area/cultivated land area. The fertilizer use intensity variable is defined as the fertilizer application amount per unit area (i.e., total fertilizer application amount/cultivated land area). The related data of the total area of household contracted cultivated land, the area of cultivated land, and the total amount of chemical fertilizer application, all come from China Rural Statistical Yearbook. The air pollution data comes from the global surface PM_2.5_ concentration measured by the Atmospheric Composition Analysis Group of Washington University, the average PM_2.5_ of each province in China is shown in Figure 1.

Data in Table 1 are information about the three key variables. The second column is the sample size; the sample size is relatively large, and as such the results are more convincing. The third column is the sample mean, the last five columns are the minimum value of the data, and four points: The first quartile, the median, the third quartile, and the maximum value. The three variables have increased in the same trend in the past 20 years. China’s urbanization process basically started after 2000. In order to promote economic development, the environment is sacrificed to a certain extent. The initial minimum value of PM_2.5_ concentration was *15.4* micrograms/cubic meter, and the current maximum value is *112.7* micrograms/cubic meter. It can be found that its increase is huge, and it must attract the attention of all mankind and strive to suppress its increase.

This paper made a complete empirical test on the collected panel data. Figure 2 shows the flow chart of the empirical steps.

### 3.2. The Relational Models of Study Variables

Equation (1) specifies the generic relational model used to illustrate the connections to investigate the relationship between the study variables. PM_2.5_ denotes air pollution, transferate denotes land transfer rate, perfertilizer denotes fertilizer use rate, *i* denote the number of Chinese provinces, and *t* defines the time in the relational model. The following is the broad relational model that relates the study variables:(1)$${\mathrm{PM}}_{2.5}=f\left({\mathrm{landtransferate}}_{it},{\mathrm{perfertilizer}}_{it}\right)$$

According to Gauss–Markov theory, the econometric form of the above model is as follows:(2)$${\mathrm{PM}}_{2.5}=\alpha +\beta_{1}land{\mathrm{transferate}}_{it}+\beta_{2}{\mathrm{perfertilizer}}_{it}+\varepsilon_{it}$$
where α is the intercept, $\beta_{1}$ and $\beta_{2}$ are explanatory variable coefficients, and $\varepsilon_{it}$ is the error term. The other variables are the same as those previously mentioned.

### 3.3. Econometric Methodology

This study investigated the causal relationship between land transfer, fertilizer use, and PM_2.5_ pollution in China. In order to help achieve the research goal, a series of tests were carried out, including a cross section test, a unit root test, a cointegration test and a Granger causality test.

#### 3.3.1. Cross Section Test

Land transfer, fertilizer usage, and PM_2.5_ pollution may be affected by the same factors, such as advances in agricultural tools. Therefore, the cross-sectional correlation of panel data must be considered-otherwise, the estimated results will be biased. In this study, we adopted the *CD* test [55] and Lagrange multiplier test (*LM* test) developed by Pesaran.

The *CD* test formula is as follows:(3)$$CD=\sqrt{\frac{2T}{N\left(N-1\right)}}\left(\overset{N-1}{\underset{i=1}{\sum }}\overset{N}{\underset{j=i+1}{\sum }}\rho_{ij}\right)$$
where *N* and *T* are cross sections and periods; $\rho_{ij}$ stands for the cross-sectional correlations of the error between *I* and *j*.

The *LM* test formula is as follows:(4)$$LM=\sqrt{\frac{1}{N\left(N-1\right)}}\overset{N-1}{\underset{i=1}{\sum }}\overset{N}{\underset{j=i+}{\sum }}\left(T_{ij}\mu_{ij}^{2}-1\right)\to N\left(0,1\right)$$

#### 3.3.2. Unit Root and Stationarity Tests

Before formally studying the relationship between the selected variables, it is necessary to confirm whether the sequence is stable after the level value or the first-order difference. In this paper, the unit root test methods, such as Levin-Lin-Chu (LLC) [56], Im-Pesaran-Shin (IPS) test [57], ADF and PP test are adopted, and the unit root test is a normative test for determining whether the sequence is an I (1) process.

##### LLC

The LLC unit root test method is based on the DF and ADF test formulas of traditional single time-series data, and considers the autocorrelation of a single individual as the disturbance term. The inspection process mainly includes the following two steps: (1) First, remove the autocorrelation and deterministic items from $\Delta y_{it}$ and$\;y_{it}$ influence and normalize it as a proxy variable. (2) Conduct ADF regression with proxy variables.

The hypothetical form of the LLC unit root test method:

**H0:** 
*ρ = 0 (has a unit root); H_1_: ρ < 0. The LLC test is a left one-ended test.*


##### Im, Pesaran, and Shin (IPS Inspection)

LLC is suitable for the same root test, and IPS is suitable for different root tests. IPS does not require the assumption that each individual is completely homogenous. This is a major feature and is superior to LLC, which can be extended to deal with unbalanced panel data.

##### ADF test

ADF is an augmented Dickey-Fuller test. Dicky and Fuller proposed an enhanced DF test, which formed the enhanced Dickey-Fuller test. The formulas for its three forms are as follows:(5)$$y_{t}=\overset{p-1}{\underset{i=1}{\sum }}\zeta_{i}\Delta y_{t-i}+\rho y_{t-1}+\varepsilon t\;\;$$
(6)$$y_{t}=\overset{p-1}{\underset{i=1}{\sum }}\zeta_{i}\Delta y_{t-i}+\alpha +\rho y_{t-1}+\varepsilon t$$
(7)$$y_{t}=\overset{p-1}{\underset{i=1}{\sum }}\zeta_{i}\Delta y_{t-i}+\alpha +\beta t+\rho y_{t-1}+\varepsilon t$$

The hypothetical form for the three forms is H_0_: |ρ| ≥ 1; H_1_: |ρ|≤ 1.

Test statistics t = $\frac{\hat{\rho }-1}{{\hat{\sigma }}_{\rho }}$

##### PP Inspection

DF test is divided into ordinary AR(1) processes: $y_{t}=\rho y_{t-1}+\varepsilon_{t}$; AR(1) process with drift term: $y_{t}=\rho y_{t-1}+\alpha +\varepsilon_{t}$; trend stationery AR(1) process: $y_{t}=\rho y_{t-1}+\alpha +\beta t+\varepsilon_{t}$ three cases. The PP test optimizes the DF statistic and corrects the DF statistic by a nonparametric method, so that it has the function of lag period estimation.

#### 3.3.3. Optimal Lag Selection

Lag selection is required for economic tests such as the unit root, stationarity, and cointegration. On the other hand, inappropriate lag selection may result in freedom-related constraints. The Akaike information criterion (AIC), sequential modified LR test statistic (L-R), Schwarz information criterion (SC), final prediction error (FPE), and Hannan-Quinn information criterion (HQ) are used to determine the appropriate delays.

#### 3.3.4. Cointegration Test

Cointegration analysis was performed after the unit roots of the series had been explored. The cointegration analysis is a method for determining whether or not two variables have a long-term relationship. The Kao cointegration analysis methods are employed in this scenario. Kao [58] created a cointegration test based on the DF (Dickey-Fuller) and ADF (Generalized Dickey-Fuller) tests. The null hypothesis in Kao’s panel cointegration test is “no cointegration,” and the alternative hypothesis is “cointegration exists.” In the relevant stage of ADF test statistics, the null hypothesis is rejected.

The model considered by Kao [59] is as follows:(8)$$y_{it}=\alpha_{i}+X_{it}^{\prime }\beta +e_{it},i=1,2,\ldots ,N;t=1,2,\ldots ,T\;\;\;\;\;\;$$
where $X_{it}$ is an m-dimensional column vector, assuming that for each individual *i*, $y_{it}$ and$\;X_{it}^{\prime }$ are all I(1) variables, under the null hypothesis of no cointegration; $e_{it}$ will also be an I(1) variable. Using Equation (9), the residual ${\hat{e}}_{it}$ of OLS regression, the auxiliary regression formula of the ADF test proposed by Kao is: (9)$$\;\;\;\;\;{\hat{e}}_{it}=\rho {\hat{e}}_{i,t-1}+\overset{p}{\underset{j=1}{\sum }}\delta_{j}\Delta {\hat{e}}_{i,t-j}+v_{it}$$

The corresponding statistic is recorded as ADF, and the statistic sequentially converges to the standard normal distribution.

#### 3.3.5. Granger Causality

Although cointegration aids in the identification of possible short- or long-run correlations, it falls short of capturing the causal links between variables. It is critical to have empirical knowledge of the causal links to reducing air pollution. Causality analyses were utilized to help empirically determine the causal linkages among the variables employed in this study [59]. Causality studies were used to determine the long- and short-term causal links between PM_2.5_ emission, land transfer rate, and fertilizer use rate. The empirical causality models depended on the relational model’s cointegration state. Multiple time-dependent causalities could be included in a relational model that shows cointegration interactions, i.e., both long- and short-run causal linkages could be present. Hurlin et al. [60] and Law et al. [61] proposed a method to apply the standard Granger causality test to panel data. The specific model is as follows:(10)$$Y_{it}=\sum_{k=1}^{p}Y^{\left(k\right)}Y_{i,t-k}+\sum_{k=1}^{p}\beta^{\left(k\right)}X_{i,t-k}+\mu \times t+v_{it\;}$$

In the above formula, p represents the lag order and is a positive integer, and $v_{it\;}$is a random error term. The null hypothesis is H_0_: for any *k*, $\beta^{\left(k\right)}\;$*= 0*, and the alternative hypothesis is H_1_: there is k such that $\beta^{\left(k\right)}\;$≠ 0. If the null hypothesis is rejected, *X* is the Granger cause of *Y*; otherwise, *X* is not the Granger cause of *Y*. The null hypothesis can be tested with the following statistics:(11)$$F\;\frac{(SSR_{2}-SSR_{1})/p}{SSR_{1}/\left(T\times N-N-2p-1\right)}$$

Among it, $SSR_{1}$ and $SSR_{2}\;$are the residual sums of squares of OLS estimates, with and without constraints, respectively. *N* is the panel data width, *T* is the time length, and *p* is the lag order.

## 4. Results

### 4.1. Results of Cross Section Correlation Test and Unit Root Test

Table 2 shows the test results of the cross section correlation. From the results, it can be seen that the original assumption that there is no dependence between the land transfer, fertilizer usage, and PM_2.5_ was rejected at the significance level of *1%*, indicating that there is a dependence. We have adopted the LLC test, IPS test, ADF test, and PP test for the unit root test to ensure the correctness of the test. The test results are recorded in Table 3. The results of the four test methods were consistent. The variables PM_2.5_, land transfer, and fertilizer usage are not stable in level value, so there are unit roots. However, after the first-order difference, the original unit root hypothesis was rejected at the significance level of *1%*, and all the variables were stationary sequences at the same confidence level. Therefore, all the variables are first-order single-integer sequences, and they have the same-order single-integer relationship with each other, which can be used for the next cointegration test.

### 4.2. The Results of Cointegration Results

Kao’s residual panel cointegration test results test whether there is a long-term equilibrium relationship between our independent and dependent variables. Table 4 shows the results of the cointegration test. The *p*-value is *0.0329*, and the zero hypothesis was rejected at the significance level of *5%*. The research results support a long-term equilibrium relationship between PM_2.5_ pollution, land transfer, and fertilizer usage, so policy-makers should consider all these factors when making long-term environmental policies.

### 4.3. Stationarity Test

In order to explore the concrete relationship among PM_2.5_ emission, fertilizer usage, and land transfer, this paper constructed a three-bit vector autoregressive model. Impulse response function analysis and variance analysis can only be carried out if the stability is satisfied. After inspection, all the root values were distributed within the unit circle (the results are shown in Figure 3), which indicate that the PVAR model constructed in this study has good stability.

### 4.4. Fully Modified Least Squares and Dynamic Least Squares Estimation Results

The detailed results of the fully modified OLS (FMOLS) and the dynamic OLS (DOLS) are shown in Table 5. The research results show that fertilizer usage was positively correlated with PM_2.5_ emissions, whereas the landtransferate was negatively correlated with PM_2.5_ emissions. In the long run, under the FOLS method, fertilizer usage increased by *1%*, PM_2.5_ emissions increased by about *0.17%*, landtransferate increased by *1%*, PM_2.5_ decreased by about *0.07%*, and, under the DOLS method, fertilizer usage increased by 1%. The emission of PM_2.5_ increased by about *0.58%*, the landtransferate increased by 1%, and the PM_2.5_ decreased by about *0.08%*. All the results were statistically significant at the level of *0.01*. The sign of the coefficients of FMOLS estimation and DOLS estimation was consistent, which indicated that our estimation was stable and reliable. Both of them indicated that the increase in chemical fertilizer use aggravated the environmental pollution and worsened the environmental situation. In contrast, the increase in landtransferate helped to alleviate the environmental problems and had a positive effect on curbing environmental pollution. These conclusions have been confirmed in the existing literature [12,50].

### 4.5. Granger Causality Test Results

The Granger causality test is used to examine the relationship between variables and measure whether the change of one variable is the cause of the change of another variable. The Granger causality test results of variables are shown in Table 6. The results showed that the amount of fertilizer use and PM_2.5_, land transfer and PM_2.5_, land transfer and fertilizer use all showed two-way causality, and all the results were significant at the *1%-10%* level, indicating that the relationship between the three core variables is closely related and they affect each other, especially the contribution of chemical fertilizer use to PM_2.5_ pollution is huge.

### 4.6. Impulse Response and Variance Decomposition

In order to measure the contribution of endogenous variables to the impact, we analyzed the variance of variables. The results are shown in Table 7. It can be seen that in the absence of any external influence, PM_2.5_ pollution had the greatest contribution to itself. With the development of time, its influence is gradually decreasing, with the variance explanation rate falling from *100%* in the first period to about *92.15%* in the 15th period. The contribution rate of other factors is on the rise, with the contribution rate of chemical fertilizer and land circulation rising from 0% in the first period to *0.95%* and *9.9%*, respectively. Similarly, during the forecast period of the 15th period, the fertilizer usage is mainly affected by itself, with its own contribution rate remaining above *96%*. Then, affected by the land transfer, the contribution rate reached the highest value in the fifteenth period, which was about *3.16%*. The effects of PM_2.5_ was not significant, and the maximum explanation rate was only *0.58%.* However, compared with the first period, the variance explanation rate of landtransferate until the 15th period of lag increased by about *13.54* percentage points, which is path dependent. The variance explanation rate of PM_2.5_ in the 15th period of lag had increased by about *0.23* percentage points, whereas that of chemical fertilizer had decreased by about *13.76* percentage points.

The impulse response function was utilized to represent the shock-induced changes in the variables. Figure 4 depicts the results of the responses in each variable.

Figure 4a,b show the impulse responses of PM_2.5_ to fertilizer use and land transfer. From the results of the effect of chemical fertilizer use on PM_2.5_ in (a), there was a time lag in the first period, a slight upward trend in the second period, a negative response after the third period, and a negative number after crossing the zero axis, and finally, close to the zero axis tended to be stable in the long term, indicating that fertilizer use increased PM_2.5_ in the short term. Judging from the impact of PM_2.5_ on the land transfer rate, there was a time lag in the first period, and the second and third periods had continuous positive responses, reaching the maximum positive response (about *0.15*), and after that, it had been a negative response. Gradually increasing, indicating that in the long run, the effects of reducing PM_2.5_ in the land transfer rate was more obvious, but in the short term, farmers prefer to use chemical fertilizer rather than land transfer to obtain economic results.

Figure 4c,d show the impulse responses of fertilizer use to PM_2.5_ and land transfer. It can be seen from (c) that after being impacted by one standard deviation from LnPM_2.5_, LNPERFERILIZER responded positively in the first two periods, then becomes negative, and began to stabilize after the 10th period and converged to about −0.01 for a long time. There is a bidirectional causal relationship between PM_2.5_ and fertilizer use. It can be seen from (d) that, after being impacted by one standard deviation from Lnlandtransferate, LNPERFERILIZER had a negative response in the first two periods, and gradually converged from the third period. Land circulation reduced the use of fertilizers to a certain extent.

Figure 4e,f are the impulse responses of land circulation to PM_2.5_ and fertilizer use. The impact of PM_2.5_ on land transfer was small, whereas land circulation had a negative response to fertilizer use throughout the impact period, and the negative response continued to increase. The impact of chemical fertilizer use on land transfer is negative.

## 5. Conclusions and Policy Implications

Environmental pollution has become an increasingly important obstacle to the development of all countries in the world. Alleviating air pollution has become an imminent concern, and is also related to the realization of a better life and the sustainable development of human beings in the future. Therefore, exploring the influencing factors and emission reduction measures of PM_2.5_ emissions is of great significance. This paper studied the causal relationship between land transfer, fertilizer usage, and PM_2.5_ pollution. It considered the influence of economic development and urbanization on carbon emissions through a whole set of empirical processes, having obtained the corresponding empirical results. 

First, the cross-sectional correlation test verified that there was a dependency between land transfer, fertilizer use, and PM_2.5_. The unit root test of the panel data was carried out using an LLC test, an IPS test, an ADF test, and a PP test to analyze the stationarity of the variables. The test results in Table 3 showed that the variables after the first-order difference were stationary, indicating that the LnPM_2.5_, Lnfertilizer, and Lnlandtransferate sequences were single-integrated sequences of the same order, and that the PVAR model can be regressed. At the same time, the results of the Kao test rejected the null hypothesis, “there is no cointegration relationship between variables”, indicating a cointegration relationship between the three variables. 

In the subsequent stationarity test of the PVAR model, it can be observed that all the reciprocals of the unit root were less than one, and the blue bullets were distributed within the unit circle. The estimation results of FMOLS and DOLS mainly showed the influence coefficient of chemical fertilizer use and land transfer on PM_2.5_. The results all showed that fertilizer use was positively correlated with PM_2.5_ emissions, but land transfer was negatively correlated with them. The Granger causality test was used to test the causal relationship between PM_2.5_, land transfer, and fertilizer use. The results showed that there was a causal relationship between the three variables, and they affected each other. The impulse response function of the PVAR model (shown in Figure 4) more intuitively reflects the dynamic interaction and effect size of fertilizer use on PM_2.5_. The conclusion is that fertilizer use increases PM_2.5_ in the short term, and variance decomposition also evaluates the contribution rate of each variable to the fluctuation of endogenous variables.

Through the demonstration of the empirical results, the validity and significance of putting chemical fertilizer use, PM_2.5_, and land circulation into the same system for research are ensured. At the same time, it ensures the correctness of the model construction, with no false regressions. FMOLS, DOLS estimation results, impulse response function, and variance decomposition reached the same conclusion: that fertilizer use increased PM_2.5_ emissions and brought environmental pollution. This is consistent with the research views and conclusions of Li et al. [62], who determined that reducing agricultural NH_3_ emissions can effectively reduce PM_2.5_ pollution [63]. Xu et al. [64] believed that PM_2.5_ was significantly positively correlated with cultivated land area, and the fragmentation of cultivated land was beneficial to the decrease of PM_2.5_. However, the correlation coefficient between the land transfer rate and PM_2.5_ emissions was negative at the 1% significance level, indicating that an increase in land transfer rate would reduce PM_2.5_ emissions. land transfer has greatly increased the land scale of large-scale farmers, has improved production efficiency [11], and has encouraged large-scale farmers to apply organic fertilizers [65], which can alleviate some environmental problems.

Based on the above research and conclusions, the following policy recommendations are put forward: (1) The government should actively promote and accelerate land transfers so that future land operations will develop on a large scale. At the same time, it should ensure the stability of land transfer quality and management rights to protect large-scale farmers’ rights and interests, encourage them to replace chemical fertilizers with organic fertilizers, and pursue the long-term interests of the land. (2) Optimize land use to actively promote ecological land improvement; improve extensive agricultural management; establish farmers’ awareness of the multidimensional balance between social, economic, and ecological benefits; and reduce PM_2.5_ pollution to improve air quality. (3) Implement a land rotation system, reduce agricultural production intensity, attach importance to the treatment and effective control of crop fertilizer residues, increase technology research and development, and use various advanced technologies to control agricultural ammonia emissions. 

This study still has some limitations, such as using the PVAR model to treat the research variables as endogenous variables, ignoring that other activities (such as industrial activities) in the sample area may also increase PM_2.5_ pollution. Considering other factors that may affect PM_2.5_ pollution is a direction for further research.

## Figures and Tables

**Figure 1 ijerph-19-08387-f001:**
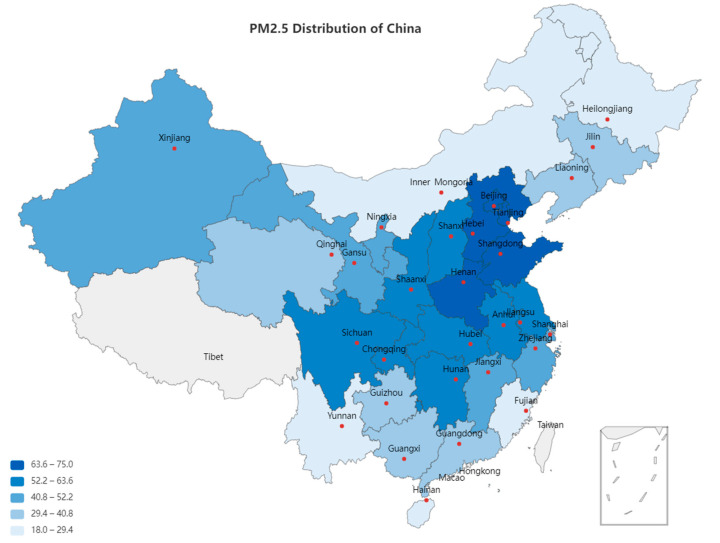
Distribution of mean PM_2.5_ of each province. Note: Data source is the Atmospheric Composition Analysis Group of Washington University.

**Figure 2 ijerph-19-08387-f002:**
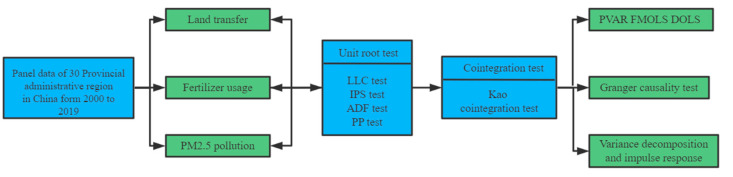
Study content and methodology framework.

**Figure 3 ijerph-19-08387-f003:**
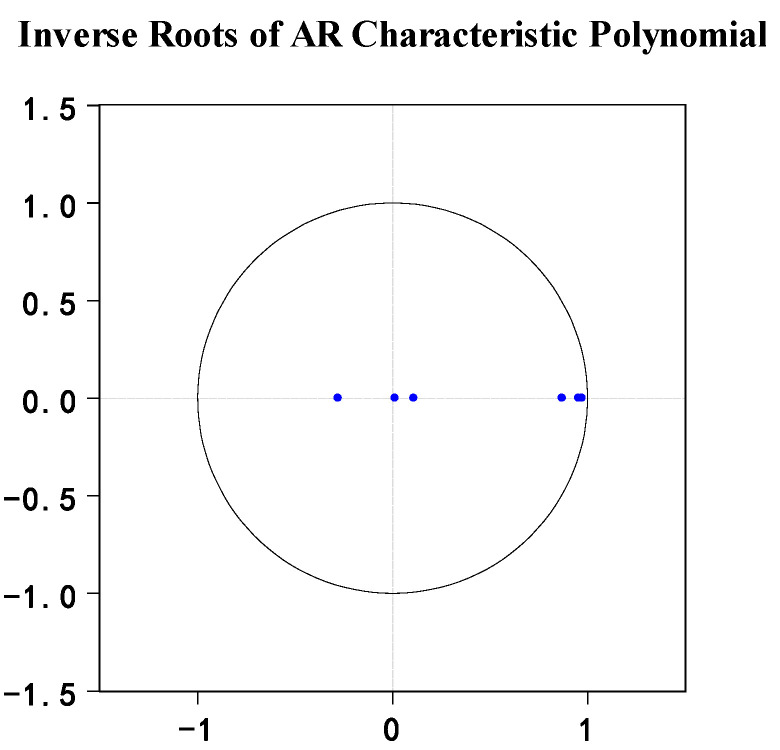
Inverse roots of the PVAR characteristic polynomial. Note: the circle in the figure represents the unit circle.

**Figure 4 ijerph-19-08387-f004:**
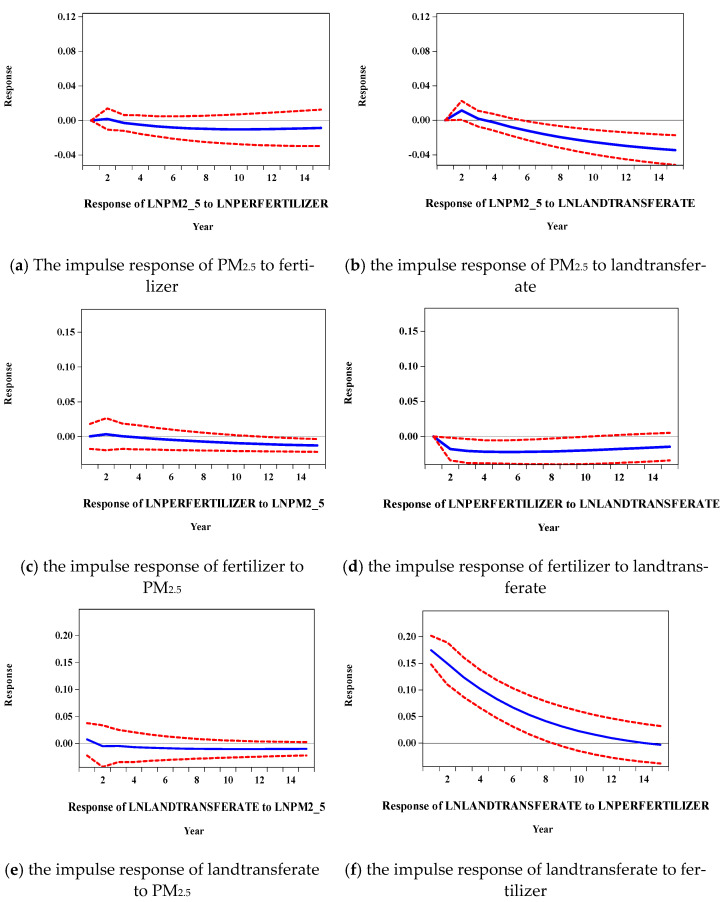
Impulse response plot for three variables. (**a**,**b**) are the graphs of the impulse response of PM_2.5_ to fertilizer and landtransferate, (**c**,**d**) are the graph of the impulse response of fertilizer to PM_2.5_ and landtransferate, (**e**,**f**) are the graph of the impulse response of landtransferate to PM_2.5_ and fertilizer. Note: red lines represent the upper and lower lines of the 95% confidence interval, blue line represents the impulse response function.

**Table 1 ijerph-19-08387-t001:** Descriptive statistics of variables of interest.

	Count	Mean	Std	Min	25%	50%	75%	Max
perfertilizer	600	0.47	0.40	0.08	0.27	0.44	0.58	4.15
landtransferate	450	0.17	0.15	0.01	0.06	0.13	0.24	0.93
PM_2.5_	570	47.28	17.18	15.40	34.50	46.80	59.55	112.70
Lnperfertilizer	600	−0.95	0.62	−2.55	−1.30	−0.82	−0.54	1.42
Lnlandtransferate	450	−2.27	1.10	−4.96	−2.90	−2.07	−1.41	−0.08
LnPM_2.5_	570	3.78	0.40	2.73	3.54	3.85	4.09	4.72

**Table 2 ijerph-19-08387-t002:** Cross-sectional dependence test results.

Test	Statistic	Probability
Breusch–Pagan LM	2756.0190	0.0000 ***
Pesaran scaled LM	78.6899	
Pesaran CD	47.3619	

Note: *** is significant at the level of 1%. The cross-sectional dependence test results in the third column of the table are all 1% significant.

**Table 3 ijerph-19-08387-t003:** Panel unit root tests results.

Variables	Level		First-Difference	
	Intercept	Intercept and Trend	Intercept	Intercept and Trend
LLC test				
LnPM_2.5_	0.0215	0.2195	0.0000 ***	0.0000 ***
Lnfertilizer	0.0000	1.0000		
Lnlandtransferate	0.2869	0.7774		
Im, Pesaran, and Shin test				
LnPM_2.5_	0.0030	0.9886		
Lnfertilizer	0.1394	1.0000		
Lnlandtransferate	0.2952	0.9454		
ADF-Fisher Chi-square test				
LnPM_2.5_	0.0001	0.4311		
Lnfertilizer	0.2152	1.0000		
Lnlandtransferate	0.7871	0.9947		
PP-Fisher Chi-square test				
LnPM_2.5_	0.0000	0.2686		
Lnfertilizer	0.0194	1.0000		
Lnlandtransferate	0.5854	0.9991		

Note: *** is significant at the level of 1%. The results of the four panel unit root tests after the first difference are the same, and they are significant at the 1% level. LLC: Levin-Lin-Chu Test.

**Table 4 ijerph-19-08387-t004:** Kao’s residual panel cointegration test results (ADF).

	Null Hypothesis	*t*-Statistics	Probability
ADF	No co-integration	−1.8403	0.0329 **

Note: ** is significant at the level of 5%. ADF: augmented dickey–fuller test.

**Table 5 ijerph-19-08387-t005:** The results of FMOLS, DOLS estimation techniques: full panel.

Variables	Coefficient	Standard Error	*t*-Statistic	Probability
FMOLS				
LNPERFERTILIZER	0.1672	0.0615	2.7184	0.0069 ***
LNLANDTRANSFERATE	−0.0723	0.0130	−5.5514	0.0000 ***
DOLS				
LNPERFERTILIZER	0.5841	0.1609	3.6297	0.0004 ***
LNLANDTRANSFERATE	−0.0869	0.0250	−3.4839	0.0007 ***

Note: *** is significant at the level of 1%. FMOLS: fully modified least squares. DOLS: dynamic least squares.

**Table 6 ijerph-19-08387-t006:** Pairwise Granger Causality Tests.

Null Hypothesis	F-Statistic	Probability
LNPERFERTILIZER is not a cause of LNPM_2.5_	10.8216	0.0000 ***
LNPM_2.5_ is not a cause of LNPERFERTILIZER	9.4480	0.0000 ***
LNLANDTRANSFERATE is not a cause of LNPM_2.5_	2.3823	0.0187 **
LNPM_2.5_ is not a cause of LNLANDTRANSFERATE	1.8407	0.0730 *
LNLANDTRANSFERATE is not a cause of LNPERFERTILIZER	1.8220	0.0750 *
LNPERFERTILIZER is not a cause of LNLANDTRANSFERATE	2.3372	0.0203 **

Note: *** is significant at the level of 1%, ** is significant at the level of 5%, and * is significant at the level of 10%.

**Table 7 ijerph-19-08387-t007:** Variance decomposition result.

Period	Standard Error	LNPM_2.5_	LNPERFERTILIZER	LNLANDTRANSFERATE
Variance Decomposition of LNPM_2.5_				
1	0.115135	100.0000	0.000000	0.000000
2	0.139743	99.30975	0.016292	0.673959
3	0.164213	99.45920	0.040039	0.500763
4	0.183580	99.48060	0.099349	0.420056
5	0.200954	99.30485	0.194925	0.500228
6	0.216554	98.95226	0.306540	0.741204
7	0.230904	98.44637	0.422994	1.130635
8	0.244235	97.82241	0.534944	1.642641
9	0.256733	97.11040	0.636664	2.252931
10	0.268522	96.33659	0.724818	2.938590
11	0.279692	95.52227	0.797946	3.679785
12	0.290312	94.68443	0.855882	4.459684
13	0.300434	93.83634	0.899323	5.264339
14	0.310101	92.98815	0.929495	6.082354
15	0.319346	92.14754	0.947918	6.904544
Variance Decomposition of LNPERFERTILIZER				
1	0.169845	8.68 × 10^−5^	99.99991	0.000000
2	0.231111	0.019958	99.37400	0.606041
3	0.271837	0.014629	98.96764	1.017731
4	0.301632	0.013770	98.63929	1.346938
5	0.324492	0.021302	98.35129	1.627407
6	0.342535	0.037971	98.08514	1.876892
7	0.357054	0.064266	97.83485	2.100885
8	0.368905	0.100043	97.59800	2.301960
9	0.378686	0.145043	97.37364	2.481319
10	0.386829	0.198853	97.16129	2.639854
11	0.393660	0.260988	96.96058	2.778436
12	0.399426	0.330908	96.77104	2.898057
13	0.404321	0.408037	96.59211	2.999853
14	0.408496	0.491780	96.42313	3.085095
15	0.412074	0.581532	96.26332	3.155150
Variance Decomposition of LNLANDTRANSFERATE:				
1	0.284353	0.074641	37.79296	62.13239
2	0.387579	0.056880	35.21509	64.72803
3	0.451648	0.052132	33.42575	66.52212
4	0.496082	0.061876	31.91942	68.01871
5	0.528335	0.076144	30.60939	69.31446
6	0.552514	0.094534	29.45579	70.44968
7	0.571050	0.115456	28.44131	71.44323
8	0.585521	0.138153	27.55349	72.30836
9	0.596999	0.161883	26.78174	73.05637
10	0.606232	0.186067	26.11587	73.69807
11	0.613759	0.210215	25.54577	74.24402
12	0.619968	0.233930	25.06150	74.70457
13	0.625151	0.256899	24.65341	75.08969
14	0.629522	0.278881	24.31230	75.40882
15	0.633245	0.299700	24.02956	75.67074

## Data Availability

Data supporting the conclusions of this article are included within the article. The datasets presented in this study are available on request from the corresponding author.

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
