# Peer review of "Assessing the Relationship among Land Transfer, Fertilizer Usage, and PM2.5 Pollution: Evidence from Rural China"

_ijerph, 2022, doi:10.3390/ijerph19148387_

Round 1
Reviewer 1 Report
1. What is the reason for the work not reaching at least 2021 (time cut from 2000 to 2019)?
2. I suggest changing the keywords. For example, "land transfer" does not mirror the work and will make it challenging to be found via metadata search. The same goes for "PVAR."
3. What was the reason for choosing the 30 provinces? Are there other activities in the locations? For example, industrial? What are the cross emission factors?
4. Was the equation developed by the authors? It seems to me that the modeling comes from the Dickey-Fuller test. In this case, is there a need to set it entirely in the article?
5. I suggest the authors increase the description of the method applied to the case they studied instead of dwelling on the description of the technique itself. The description of the issue was confusing.
6. The curves in Figure 3 present vital data, however, they are too small to be read
7. The conclusions could be better specified. Going into data on government policies without having specified them in the body of the article strikes me as disconnected text. I suggest reflection on the conclusions and rewriting.
Author Response
First of all, we would like to thank Reviewer 1 for reading our article and for your valuable comments. Below is our response to all comments made.
Point1. What is the reason for the work not reaching at least 2021(time cut from 2000 to 2019)?
Response 1:We thank the reviewer for raising this question. We know that using the data of 2021 can better investigate the relationship between the three variables, thus making the results more convincing. But so far, China Rural Statistical Yearbook does not disclose data for 2020 and beyond. Fortunately, this does not affect the overall study of the relationship between land transfer, fertilizer usage and PM2.5 pollution, because the 20-year data available are enough for us to use PVAR and other methods to explore the relationship between them.
Point2. I suggest changing the keywords. For example, "land transfer" does not mirror the work and will make it challenging to be found via metadata search. The same goes for "PVAR."
Response 2: High-quality keywords are the key to enabling papers to be retrieved by needed readers via metadata search. To make the selected keywords mirror the work more, we decided to delete the keyword PVAR, but still retain the land transfer. The reason for retaining land transfer is that land transfer is a very important concept in China. Only through the process of rural land transfer can China's agriculture be mechanized and intensive. Many scholars have conducted various studies on land transfer in China and used it as the paper keyword[1–3].
- Xi, Q.; Mei, L. How Did Development Zones Affect China’s Land Transfers? The Scale, Marketization, and Resource Allocation Effect. Land Use Policy 2022, 119, 106181.
- Wang, B.; Zhang, Y.; Feng, S. Impact of the Low-Carbon City Pilot Project on China’s Land Transfers in High Energy-Consuming Industries. Journal of Cleaner Production 2022, 363, 132491.
- Tan, J.; Cai, D.; Han, K.; Zhou, K. Understanding Peasant Household’s Land Transfer Decision-Making: A Perspective of Financial Literacy. Land Use Policy 2022, 119, 106189.
Point3: What was the reason for choosing the 30 provinces? Are there other activities in the locations? For example, industrial? What are the cross emission factors?
Response3: (1) The purpose of this paper is to explore the relationship between rural land transfer, fertilizer usage and PM2.5 pollution in China, so we intend to use as many provincial administrative regions as possible, but considering the great differences in data statistical calibre and geographical environment, we exclude Tibet, Hong Kong, Macao and Taiwan, and use the remaining 30 provincial administrative regions as samples.
(2) In line with the examples of reviewers, there are industrial activities in all provinces of China. Industrial activities and other factors are also important causes of PM2.5 pollution, but one of the advantages of the PVAR model used in this paper is to take all the variables to be studied as endogenous variables and explore the influence of one endogenous variable on other variables, so the influence of industrial activities can be ignored. However, the interference factors such as industrial activities put forward by the reviewer are the direction that we continue to study. We added this sentence in the penultimate paragraph:’’ However, this study still has some limitations, such as using the PVAR model to treat the research variables as endogenous variables, ignoring that other activities such as industrial activities in the sample area may also increase PM2.5 pollution. Considering other factors affecting PM2.5 pollution is the direction of further research’’, and hope that we or other researchers can continue to study.
Point4: Was the equation developed by the authors? It seems to me that the modelling comes from the Dickey-Fuller test. In this case, is there a need to set it entirely in the article?
Response4: As the reviewer said, this equation comes from the Dickey-Fuller test, and we follow the reviewer's suggestion to cut this part to the following:
ADF is an augmented Dickey-Fuller test. The formulas for its three forms are as follows:
Point5: I suggest the authors increase the description of the method applied to the case they studied instead of dwelling on the description of the technique itself. The description of the issue was confusing.
Response5: For some commonly used methods, it is important to introduce how to apply this method to the content of the study rather than a lengthy introduction technique itself. We accept the reviewer’s suggestion, delete the introduction of methods that are not closely related to the paper, and retain only the basic meaning. And we have added the description of the method applied to the case, such as Land transfer, fertilizer usage and PM2.5 pollution may be affected by the same factors, such as advances in agricultural tools. Therefore, the cross-sectional correlation of panel data must be considered, otherwise, the estimated results will be biased.
Point6: The curves in Figure 3 present vital data, however, they are too small to be read
Response: It is caused by our carelessness. We have replaced a better picture.
Point7: The conclusions could be better specified. Going into data on government policies without having specified them in the body of the article strikes me as disconnected text. I suggest reflection on the conclusions and rewriting.
Response7: Maybe our previous description misled the reviewer. We have changed the misleading “city” into “provincial administrative region”. The sample data of this paper are from 30 provincial administrative regions in China, which objectively reflects the current situation of PM2.5 pollution, land transfer and fertilizer use in China, rather than data on government policies. The purpose of our study is to find the relationship between variables through 20 years of statistical data, which could better advise the government. However, to be more rigorous and easier to understand, we have revised the conclusion.

Reviewer 2 Report
Highlight changes in yellow in a next revision, please. No track changes.
Dear authors, the similarity in the manuscript should be addressed, particularly because already in the abstract similar content appears.
Please clarify all abbreviations used first
Which are?
“The key variables of this paper”
I would like to see quantitative data in the abstract.
Despite using latest references, please also assure they that they are international enough.
Please specify that you are presenting the research hypothesis.
“H1: Land transfer has a significant negative impact on the use of chemical fertilizers.”
Then again.
“H2: The use of chemical fertilizers has a significant positive effect on the concentration 170
of PM2.5.”
and more…
to divide the hypothesis through the text is unclear to me. The reader will not focus onto the main aims of this text.
There are other ways to do this.
“about 200 tons (1 ton = 1012 grams)
Please edit as a reference and include all these cases in the final list of references too.
“output (www.fao.org/faostat/)”
Please address all subscript.
“High Ammonia (NH3)”
There seems to be an inconsistency and already defined differently above.
“Ammonia (NH3)”
Why start with a lower letter? And please do not start hating this with the word “The”.
“2.3. the land transfer and PM 2.5 pollution”
This kind of statements must be carefully revised because it is not clear to what region do you refer to.
“More and more farmers are engaged in land transfer (leasing other farmers' land) to 173
expand the scale of farmland they manage”
See that all captions must be self-explanatory by their own. Also, the picture quality is not good enough.
“Figure 1. Flowchart of the methodology.”
Try to be more clear in headings
“3.2. The relational models”
Please assure that all equations without the proper reference before are completely original, and then highlight that. Not really. Usually the case. Also check the format of the parameters being defined in the text and then include units wherever available.
Please limit known data used in the text, otherwise the manuscript is transformed into a mathematical text.
See that reference number must be present all over.
“The model considered by Kao is as follows:”
In tables. Please define all abbreviations under the table. There is no point in having an entire column in the table with the same information. In these cases that can be added under a note. This is the example of column 3
“Table 1. Cross-sectional dependence test results.”
Please check all other cases.
There are many examples of italics to be addressed in the text when relating to parameters.
Why repeat in the caption? The information above in the title of the graphic?
“Figure 2. The inverse roots of the AR characteristic polynomial.”
Please avoid using abbreviations in headings.
“4.4. FMOLS and DOLS estimation results
The same in captions.
“Table 4. The results of FMOLS, DOLS estimation techniques: full panel””
There needs to be other way to represent the variables in the tables as the null hypothesis…
“LNLANDTRANSFERATE does not Granger Cause LNPERFERTI
LIZER”
Despite the obvious interest of this manuscript. I believe that the authors should find a way to easy the way information is presented to the readers.
This figure is just unreadable. Also, all figures are grouped, so in this case a letter must correspond to each figure and the sub caption must be added after the main caption.
“Figure 3. Impulse Response Plot for Three Variables”
Before this statement, the authors could try to defend the paper with a brief contextualization. Does justifying the need to publish this paper.
“5. Conclusions and Policy Implications 466
This paper studies the causal relationship between land transfer, fertilizer usage and 467
PM2.5 pollution in China from 2000 to 2019. According to the existing literature, this is the 468
first time to study the relationship between the three.”
As immediately seen the manuscript needs additional entire proofreading.
“Given this study, we put forward 479
the following policy suggestions: 1. The government actively promotes land circulation, 480
ensures the quality of land circulation and the stability of management rights, encourages 481
large-scale farmers to use organic fertilizers, reduce synthetic fertilizer usage, and pursues 482
the long-term interests of land; 2. Optimize land use, actively promote ecological land 483
consolidation, improve extensive agricultural management, enhance farmers' awareness 484
of social, economic and ecological multi-dimensional benefits, reduce PM2.5 pollution and 485
improve air quality; 3. Implement the rotation system, reduce the intensity of agricultural 486
production, pay attention to the treatment and effective control of crop stubble and ferti- 487
lizer residue, and use various advanced technologies to control the emission of agricul- 488
tural ammonia. 48”
define means are not clear enough in the conclusions. As the practical implications.
“The empirical results 474
of this study show several critical findings of the factors studied: there is a significant 475
positive correlation between chemical fertilizer use and PM2.5 pollution, and chemical fer- 476
tilizer use increases air pollution. The correlation coefficient between land transfer rate 477
and PM2.5 emission is negative at the significant level of 1%, which indicates that the in- 478
crease in land transfer rate will reduce PM2.5 emission”
I hope the authors are able to improve the text based on the suggestions given above.
Author Response
First of all, we would like to thank Reviewer 2 for reading our article and for your valuable comments. Below is our response to all comments made.
Highlight changes in yellow in the next revision, please. No track changes.
OK
Point 1: Dear authors, the similarity in the manuscript should be addressed,
particularly because already in the abstract similar content appears.
Response 1: We accept the suggestion of the reviewers and have deleted the repeated parts in the paper.
Point 2: Please clarify all abbreviations used first
Which are?
“The key variables of this paper”
Response 2:
- Good abbreviations should function to facilitate the reader's reading understanding of the paper without being blinded. We have added full names to the first provincial acronyms that appear in each section. Panel Vector Autoregression (PVAR); the Communist Party of China (CPC); Dynamic OLS (DOLS); Fully Modified OLS (FMOLS)
- We have modified the original text to replace this vague expression. The revised content is as follows: The study on the causal relationship between land transfer, fertilizer usage and PM2.5 pollution in this paper is helpful to explore environmental change.
Point 3: I would like to see quantitative data in the abstract.
Response 3: We have made some adjustments to the description in the abstract. The results show that the use of chemical fertilizer has a significant positive impact on PM2.5 pollution, but the impact of the land transfer on PM2.5 pollution is negative. And land transfer can reduce the use of chemical fertilizer through economies of scale, thus reducing air pollution. More specifically, for every 1% increase in fertilizer usage, PM2.5 will increase by 0.17%; for every 1% increase in land transfer rate, PM2.5 will decrease by about 0.07%.
Point 4: Despite using the latest references, please also assure them that they are international enough.
Response 4: We accept the reviewer's comments, and have replaced and deleted several references that are not very international in the article.
Point 5: Please specify that you are presenting the research hypothesis.
“H1: Land transfer has a significant negative impact on the use of chemical fertilizers.”
Then again.
“H2: The use of chemical fertilizers has a significant positive effect on the concentration 170
of PM2.5.”
and more…
to divide the hypothesis through the text is unclear to me. The reader will not focus on the main aims of this text.
Response 5: It is very important for us to focus readers on the main aims of the paper. We have revised the location of the research hypothesis so that reviewers and readers can better focus on the aim of the paper.
Point 6: There are other ways to do this.
“about 200 tons (1 ton = 1012 grams)
Response 6: We have changed " about 200 tons (1 ton = 1012 grams)" to " about 200 tons (202400 grams)".
Point 7:Please edit as a reference and include all these cases in the final list of references too.
“output (www.fao.org/faostat/)”
Response 7: We accept this suggestion and have added the case to the reference list.
Point 8: Please address all subscripts.
“High Ammonia (NH3)”
Response 8: We accept this proposal and have regulated all subscripts.
Point 9: There seems to be inconsistency and already defined differently above.
“Ammonia (NH3)”
Response 9: What we need to explain is that NH3 is the chemical symbol of ammonia, and the word "high" above describes that it contains a lot of ammonia.
Point 10: Why start with a lower letter? And please do not start hating this with the word “The”.
“2.3. the land transfer and PM 2.5 pollution”
Response 10: We accepted the reviewer’s suggestion and corrected similar mistakes in the paper.
Point 11: This kind of statement must be carefully revised because it is not clear to what region you refer to.
“More and more farmers are engaged in land transfer (leasing other farmers' land) to 173
expand the scale of farmland they manage”
Response 11: We accept this proposal, and to express rigour, we have increased time and geographical restrictions. The revised sentence is as follows:
In the past 20 years, more and more farmers are engaged in land transfer (leasing other farmers' land) to expand the scale of farmland they manage in China
Point 12: See that all captions must be self-explanatory on their own. Also, the picture quality is not good enough.
“Figure 1. Flowchart of the methodology.”
Response 12: Easy-to-understand captions and clear pictures are important for a paper, and we have improved the titles and pictures. The revised caption is as follows: Study content and methodology framework
Point 13: Try to be more clear in headings
“3.2. The relational models”
Response 13: We accept this proposal, and the revised heading is as follows: The relational models of study variables.
Point 14: Please assure that all equations without the proper reference before are completely original, and then highlight that. Not really. Usually the case. Also, check the format of the parameters being defined in the text and then include units wherever available. Please limit known data used in the text, otherwise, the manuscript is transformed into a mathematical text.
Response 14: The equations used in the method part of the paper have been deduced and proved by predecessors. We adopted them directly, not original, and the reason why we did not list references is that we believe that they are fixed and general, widely circulated, and at the same time, they have been continuously improved, and may not require initial research on them. Additionally, based on the reviewer's comments, we have simplified and combined the methods and formulas used in the article without compromising the correctness of the equations.
Point 15: See that reference number must be present all over.“The model considered by Kao is as follows:”
Response 15: Ok, we have added the reference number in the specified position in the article.
Point 16: In tables. Please define all abbreviations under the table. There is no point in having an entire column in the table with the same information. In these cases that can be added under a note. This is the example of column 3 “Table 1. Cross-sectional dependence test results.” Please check all other cases.
Response 16: We accept the reviewer's comments and have changed the abbreviations in the table to full names as much as possible, and the abbreviations that have not been changed to full names have also been explained below the table
Point 17: There are many examples of italics to be addressed in the text when relating to parameters.
Response 17: Ok, we've changed the parameter to italics.
Point 18:Why repeat in the caption? The information above in the title of the graphic? “Figure 2. The inverse roots of the AR characteristic polynomial.” Please avoid using abbreviations in headings.“4.4. FMOLS and DOLS estimation results
Response 18: The reasons for repeating the chart information in the title: one is to express emphasis, and the other is that the original title and the title of the chart result are corresponding. We have changed the abbreviation in the title to the full name, which is marked in the text.
Point 19:The same in captions. “Table 4. The results of FMOLS, DOLS estimation techniques: full panel”
Response 19: Ok, we've changed the abbreviations of table 4 headings to their full names.
Point 20:There needs to be another way to represent the variables in the tables as the null hypothesis… “LNLANDTRANSFERATE does not Granger Cause LNPERFERTILIZER”
Response 20: We changed the variables in Table 5 to the null hypothesis of "LNPERFERTILIZER is not a cause of LNPM2.5"......
Point 21:Despite the obvious interest in this manuscript. I believe that the authors should find a way to ease the way information is presented to the readers.
Response 21: We very much agree with the reviewer's comments. We have simplified some of the content (such as the methods section), and divided each part more clearly (such as discussion, conclusions, and policy recommendations), we believe readers can easily get their own content of interest.
Point 22:This figure is just unreadable. Also, all figures are grouped, so in this case a letter must correspond to each figure and the sub caption must be added after the main caption. “Figure 3. Impulse Response Plot for Three Variables”
Response 22: We've reworked the impulse response plots to make them clearer, and have grouped all the plots as you suggested, one letter per plot, and added subtitles after the main title.
Point 23:Before this statement, the authors could try to defend the paper with a brief contextualization. Does justifying the need to publish this paper. “5. Conclusions and Policy Implications 466 This paper studies the causal relationship between land transfer, fertilizer usage and 467 PM2.5 pollutions in China from 2000 to 2019. According to the existing literature, this is Point 24:the 468 first time to study the relationship between the three.”
Response 23: The reviewer's suggestion is reasonable. Adding a brief context to the conclusion to defend the paper can give readers a better understanding of the significance and importance of this research. Therefore, we add the following context: “Environmental pollution has become more and more It has become an important obstacle to the development of all countries in the world. It is imminent to alleviate air pollution, and it is also related to the realization of a better life and sustainable development of human beings in the future. Therefore, it is of great significance to explore the influencing factors and emission reduction measures of PM2.5 emissions.”

Reviewer 3 Report
The topic is interesting and may be worthy of research. Nevertheless the paper has several shortcomings and the text is quite confusing.
- The source of data is not detailed. The authors only indicate China Rural Statistical Yearbook for chemical fertilizar (include reference), PM2.5 concentration measured by the Atmospheric Composition Analysis Group of Washington University (include reference) and 30 cities (which are they? include a map and why those 30 cities?).
- The results and scientific discussion are not clearly presented. This section should be split into two sections. Results must be presented in a objective way first. The results must be discussed after and comment if they corroborate, oppose or improve the one from previous or similar studies. A more self criticism approach is also missing highlighting the limitations of the work carried out in future lines of research.
- The conclusions should focus on the main results of the research in a summarized way so that readers can assess their interest in the paper. Policy implications may be transferred to discussion section.
- Bibliographic citations must respect journal standards.
Author Response
First, we would like to thank Reviewer 1 for reading our article and for your valuable comments. Below is our response to all comments made.
Point 1:The topic is interesting and may be worthy of research. Nevertheless the paper has several shortcomings and the text is quite confusing.
Response 1: We are very grateful to the reviewer for your affirmation of the theme of the article. We have revised the shortcomings and text of the article.
Point 2:The source of data is not detailed. The authors only indicate China Rural Statistical Yearbook for chemical fertilizar (include reference), PM2.5 concentration measured by the Atmospheric Composition Analysis Group of Washington University (include reference) and 30 cities (which are they? include a map and why those 30 cities?).
Response 2: The data used for the three key variables in this study are all basic data, not synthetic data. They can be obtained directly in the yearbook and the research team's calculation. Regarding the calculation of the ratio, the conventional achievement is the total amount divided by the area. Therefore, the data The source is clear. The 30 provinces are other provincial-level administrative units except Hong Kong, Macao, Taiwan and Tibet. Because of differences in statistical calibers, we do not include Hong Kong, Macao and Taiwan. Due to factors such as economy, education, and population in Tibet, the previous data are missing, so our selected data region also does not include Tibet.We have added a description in the data selection section of the article. In addition, we also draw a map of data selection regions and add descriptive statistics.
Point 3:The results and scientific discussion are not clearly presented. This section should be split into two sections. Results must be presented in a objective way first. The results must be discussed after and comment if they corroborate, oppose or improve the one from previous or similar studies. A more self criticism approach is also missing highlighting the limitations of the work carried out in future lines of research.
Response 3: We acknowledge and accept the reviewer's suggestion and have added a discussion of the results to the appropriate place in the article, also pointing out the limitations of the article.
Point 4:The conclusions should focus on the main results of the research in a summarized way so that readers can assess their interest in the paper. Policy implications may be transferred to discussion section.
Response 4:Based on the comments of the reviewer, we present the conclusions in a more summary way, while separating the conclusions and policy implications, so that readers can more clearly see the parts they are more interested in.
Point 5:Bibliographic citations must respect journal standards.
Response 5: We are very sorry to make our negligence, we have revised the bibliographic citation format of the full text according to the journal standard.

Round 2
Reviewer 1 Report
I believe that the justifications and changes made by the authors made the article fit to be published.
Congratulations to the authors and success in your work.
Author Response
many thanks to the reviewer, and we have revised the language.
Reviewer 2 Report
Highlight changes in yellow in a next revision, please. No track changes.
Please see that the hypothesis is already being justified… Language should be djusted:
“For the relationship between land transfer and fertilizer usage, we put forward the 110
following assumption: 111
H1: Land transfer has a significant negative impact on the use of chemical fertilizers. 112
The reasons are as follows:”
Dear authors, what I mean is that there is no need to translate tonnes in grammes.
“Point 6: There are other ways to do this.
“about 200 tons (1 ton = 1012 grams)
Response 6: We have changed " about 200 tons (1 ton = 1012 grams)" to " about 200 tons (202400 grams)"”
What I mean is that there are some definitions of the abbreviations and then they are repeated again.
What is the difference then…
“High Ammonia (NH3)”
“Ammonia (NH3)”
“”Point 9: There seems to be inconsistency and already defined differently above.
"Ammonia (NH3)"
As mentioned in the last revision, please do not start captions by “the” .IT is not used.
“Figure 1: The”
Dear authors,
I cannot agree with this usually known mathematical work is used to further developments, so please cite the original references.
“Point 14: Please ensure that all equations without the proper reference before are completely original, and then highlight that. Not really. Usually, the case. Also, check the format of the parameters defined in the text and include units wherever available. Please limit known data used in the text. Otherwise, the manuscript is transformed into a mathematical text.
Response 14: The equations used in the method part of the paper have been deduced and proved by predecessors. We adopted them directly, not original, and we did not list references because we believe that they are fixed and general, widely circulated, and at the same time, they have been continuously improved and may not require initial research on them. Additionally, based on the reviewer's comments, we have simplified and combined the methods and formulas used in the article without compromising the correctness of the equations.”
This is a mandatory information so the readers will be able to assess the changes made to the original formulation.
The quality of the figure is very low. Text is not easily seen.
“Figure 2. Study content and methodology framework.”
This is another example of a table which has a column in the column. Repeats the same information in every cell. In this case is authors should add this note under the table or in the caption.
“Table 2. Cross-sectional dependence test results.”
Same in other cases.
Previously…
“In tables. Please define all abbreviations under the table. There is no point in having an entire column in the table with the same information. In these cases that can be added under a note. This is the example of column 3
“Table 1. Cross-sectional dependence test results.””
There are many examples of italics to be addressed in the text when relating to parameters.
Still present all over the text.
“Response 17: Ok, we've changed the parameter to italics.”
Where are the legends to each axis then?
“Figure 3. The inverse roots of the PVAR characteristic polynomial.”
Authors seem to have not understood my comment in case of group figures and see that content cannot be read, there needs to be a different letter corresponding to each figure and then a subscription to that letter must be present after. The main caption of the figure.
“Figure 4. Impulse Response Plot for Three Variables.”
Previously…
“This figure is just unreadable. Also, all figures are grouped, so in this case a letter must correspond to each figure and the sub caption must be added after the main caption.
“Figure 3. Impulse Response Plot for Three Variables””
Again, because the discussion section is so tiny, you should consider merging it to the to the results section.
Please remove the headings in the conclusions. They are not necessary at all since they are addressed in the main heading of this section.
“6.1.Conclusions”
“6.2.Policy Implications”
Please do not use personal language such as we our headset into text. It is not used in internationally scientific journals.
I acknowledged the improvement in demand script, but I believe the above comment should still be addressed, particularly the formulation references.
There are still many formal corrections that need to be done.
I do not see great additions in terms of international references in the text. What I mean is that if the reality focused is the Chinese one, authors should not only include Chinese or Chinese region or regional references.
Did quality and scope of the references will assist the authors to reach a more international audience, making this study more relevant at international level. Consider that.
Again, I hope the authors are able to improve the text based on the suggestions given above.
Author Response
Response to Reviewer 2 Comments
Point1:Please, see that the hypothesis is already justified… Language should be adjusted: “For the relationship between land transfer and fertilizer usage, we put forward the 110 following assumption: 111
H1: Land transfer has a significant negative impact on the use of chemical fertilizers. 112 The reasons are as follows:”
Response1: We accept the suggestion of the reviewer. We have adjusted the language that describes the hypothesis. However, we still have to explain why the hypothesis is put forward. Although the hypothesis has been justified by other scholars, it is reasonable to put forward the hypothesis before the empirical test, considering the differences in data sources, time range and empirical methods.
“For the relationship between land transfer and fertilizer usage, we put forward the following assumption:
H1: Land transfer has a significant negative impact on the use of chemical fertilizers.
The basis for our hypothesis is as follows”
Point2: Dear authors, what I mean is that there is no need to translate tonnes in gramme
Response2: I'm sorry we misunderstood the reviewer's opinion. And we have modified it correctly.
Point3: What I mean is that there are some definitions of the abbreviations and then they are repeated again.
What is the difference then…
“High Ammonia (NH3)”
“Ammonia (NH3)”
There seems to be inconsistency and already defined differently above. "Ammonia (NH3)"
Response3: Dear reviewer, I am sorry for the misunderstanding caused by our unclear description. We have adjusted the abbreviations according to your suggestion.
“Ammonia, the chemical formula is NH3. High concentration of NH3 from agriculture contributes significantly to PM2.5 pollution in China, and fertilization is the most crucial agricultural source of NH3 and PM in the atmosphere.”
Point4: As mentioned in the last revision, please do not start captions with “the” .It is not used. “Figure 1: The”
Response4: I'm sorry it hasn't all been revised. We revised the captions of the manuscript again.
Point5:Dear authors,
I cannot agree with this usually known mathematical work is used to further developments, so please cite the original references.
……
This is mandatory information so the readers will be able to assess the changes made to the original formulation.
Response5: Ok, I'm very sorry, it was our problem before, we accept the reviewer's suggestion, and have written and marked the source or the author of the equation in the article.
Point6: The quality of the figure is very low. Text is not easily seen.
“Figure 2. Study content and methodology framework.”
Respons6: We accept the suggestion of the reviewer. We replaced it with a vector picture in SVG format. If the text is not clear, you can enlarge the image to see it.
Point7: This is another example of a table which has a column in the column. Repeats the same information in every cell. In this case, is authors should add this note under the table or in the caption.
“Table 2. Cross-sectional dependence test results.” Same in other cases.
Previously…
“In tables. Please define all abbreviations under the table. There is no point in having an entire column in the table with the same information. In these cases that can be added under a note. This is the example of column 3 “Table 1. Cross-sectional dependence test results.”
Response7: OK, as suggested by the reviewers, we have combined the columns with the same information in the table and commented below the table.
Point8: There are many examples of italics to be addressed in the text when relating to parameters.
Still present all over the text.
Response8: We are very sorry, it was our negligence, we have re-italicized the parameters in the text.
Point9: Where are the legends to each axis then?
“Figure 3. The inverse roots of the PVAR characteristic polynomial.”
Response9: We accept the suggestion of the reviewer. To make it easier for readers to understand the meaning of this picture, we have added a note “the circle in the figure represents the unit circle”.
Point10: Authors seem to have not understood my comment in the case of group figures and see that content cannot be read, there needs to be a different letter corresponding to each figure and then a subscription to that letter must be present after. The main caption of the figure.
“Figure 4. Impulse Response Plot for Three Variables.”
Response10: We have accepted the reviewer's suggestion and have replaced the impulse response plots with a separate letter for each plot, and have made changes in the explanation accordingly.
Point11: Again, because the discussion section is so tiny, you should consider merging it to the results section.
Response11: We have followed the reviewer's advice and have incorporated the Discussion and Conclusions.
Point12: Please remove the headings in the conclusions. They are not necessary at all since they are addressed in the main heading of this section.
“6.1.Conclusions”
“6.2.Policy Implications”
Response12: OK, we've removed the title from the conclusion.
Point13: Please do not use personal language such as our headset in text. It is not used in international scientific journals.
Response13: Ok, the reviewer's suggestion is very reasonable, we fully accept it.
Point14: I acknowledged the improvement in the demand script, but I believe the above comment should still be addressed, particularly the formulation references.
Response14: Ok, we worked hard to resolve the above comment based on your suggestion.
Point15: There are still many formal corrections that need to be done.
Response15: Thanks for the reviewer's suggestion, we have revised some places. If we have not revised or found something, I hope the reviewer can put it forward, and we will do better next time.
Point16: I do not see great additions in terms of international references in the text. What I mean is that if the reality focused is the Chinese one, authors should not only include Chinese or Chinese region or regional references.
Did quality and scope of the references will assist the authors to reach a more international audience, making this study more relevant at the international level. Consider that.
Response16: Yes, the quality and scope of the references are important, and the reviewer's suggestion is forward-looking. However, the key variable studied in this paper is land transfer, which is a unique phenomenon of China's rural reform. And for the accuracy of citation, we may choose not very international Literature. To get more international attention, we have added the following literature when describing fertilizer use and PM2.5 [1,2]
- Kang, Y.-G.; Lee, J.-H.; Chun, J.-H.; Yun, Y.-U.; Atef Hatamleh, A.; Al-Dosary, M.A.; Al-Wasel, Y.A.; Lee, K.-S.; Oh, T.-K. Influence of Individual and Co-Application of Organic and Inorganic Fertilizer on NH3 Volatilization and Soil Quality. Journal of King Saud University - Science 2022, 34, 102068.
- Kawashima, H.; Yoshida, O.; Joy, K.S.; Raju, R.A.; Islam, K.N.; Jeba, F.; Salam, A. Sources Identification of Ammonium in PM2.5 during Monsoon Season in Dhaka, Bangladesh. Science of The Total Environment 2022, 838, 156433.
Point17: Again, I hope the authors are able to improve the text based on the suggestions given above.
Response17: Really thank the reviewer for your valuable comments on our article. We have revised it again according to your suggestions. I hope you are satisfied. If we still have problems or have done something bad, we are very sorry for this. I also hope that the reviewer can propose that we will continue to revise carefully and strive to make our articles better.